

# A zero power warming chamber for investigating plant responses to rising temperature

Keith F. Lewin, Andrew McMahon, Kim S. Ely, Shawn P. Serbin, Alistair Rogers

Environmental & Climate Sciences Department, Brookhaven National Laboratory, Upton, NY 11973, USA

*Correspondence to*: Alistair Rogers (arogers@bnl.gov)

**Abstract**

Advances in understanding and model representation of plant and ecosystem responses to rising temperature have typically required temperature manipulation of research plots, particularly when considering warming scenarios that exceed current climate envelopes. In remote or logistically challenging locations, passive warming using solar radiation is often the only viable approach for temperature manipulation. However, current passive warming approaches are only able to elevate the mean daily air temperature by ~1.5°C. Motivated by our need to understand temperature acclimation in the Arctic, where warming has been markedly greater than the global average and where future warming is projected to be ~2–3 °C by the middle of the century; we have developed an alternative approach to passive warming. Our Zero Power Warming (ZPW) chamber requires no electrical power for fully autonomous operation. It uses a novel system of internal and external heat exchangers that allow differential actuation of pistons in coupled cylinders to control chamber venting. This enables the ZPW chamber venting to respond to the difference between the external and internal air temperatures, thereby increasing the potential for warming and eliminating the risk of overheating. On the coastal tundra of northern Alaska our ZPW chamber was able to elevate the mean daily air temperature 2.6 °C above ambient, double the warming achieved by an adjacent passively warmed control chamber that lacked our hydraulic system. We describe the construction, evaluation and performance of our ZPW chamber and discuss the impact of potential artefacts associated with the design and its operation on the Arctic tundra. The approach we describe is highly flexible and tuneable enabling customization for use in many different environments where significantly greater temperature manipulation than that possible with existing passive warming approaches is desired.





## 1 Introduction

Driven primarily by rising atmospheric carbon dioxide concentration, the global mean temperature has risen by 0.85°C since the beginning of the industrial revolution (IPCC, 2013). Terrestrial ecosystems are currently limiting the rate at which our planet is warming by absorbing approximately one third of our $CO_2$ emissions, but the fate of this terrestrial

carbon sink is critically uncertain (Friedlingstein et al., 2014; Lovenduski and Bonan, 2017). A key part of reducing this uncertainty is improving understanding and model representation of the processes that underlie the response and acclimation of plants and ecosystems to rising temperature (Gregory et al., 2009; Smith and Dukes, 2013; Busch, 2015; Lombardozzi et al., 2015; Kattge and Knorr, 2007). Gaining this understanding requires the study of plant and ecosystem processes at elevated temperatures, including warming scenarios that exceed current observations (Kayler

et al., 2015; Cavaleri et al., 2015).

A number of experimental approaches have been developed to study the effects of elevated temperature including; space or elevation for time approaches (Elmendorf et al., 2015), passive warming with open-top chambers (Marion et al., 1997; Natali et al., 2014), terracosms (Phillips et al., 2011), active warming with open-topped chambers (Norby et

al., 1997), soil warming (Hanson et al., 2011; Natali et al., 2014; Peterjohn et al., 1993), infra-red lamps (Kimball and Conley, 2009; Ruiz-Vera et al., 2015; Fay et al., 2011) and large scale above and below ground warming chambers (Bronson et al., 2009; Hanson et al., 2017; Barton et al., 2010). These different approaches for manipulating temperature have clear trade-offs (Amthor et al., 2010; Aronson and McNulty, 2009; Elmendorf et al., 2015). In addition, in many regions where critical uncertainty exists, such as high and low latitudes, and in challenging locations

such as high altitude or wetlands ecosystems, logistical limitations make many of these approaches impractical. In these locations, passive warming is often the only option for temperature manipulation. Enclosures relying on solar radiation for warming must either be open-topped or have some mechanism for temperature modulation to reduce treatment variability and avoid high temperature excursions that could result in plant mortality (Aronson and McNulty, 2009; Marion et al., 1997; Wookey et al., 1993). Unattended temperature modulation of passively warmed enclosures

can be easily accomplished using currently available electronic control systems in areas with reliable electric power, but is not possible without it. Open-topped passive warming enclosures have been used previously and work well at elevating mean daily temperatures by ~1.5°C (Elmendorf et al., 2012, 2015; Marion et al., 1997). However, projected global temperature increases expected for the middle to end of this century will exceed the temperature elevation achievable with open-topped passively warmed enclosures. This is especially true in the Arctic, where current

warming is almost double the global average and where projections for the worst case emission scenarios include temperature increases of up to and beyond 7.5°C by 2100 (IPCC, 2013; Kaufman et al., 2009; Melillo et al., 2014).

In order to gain critical process knowledge of plant and ecosystem responses to the higher temperatures projected for the end of the century it is necessary to conduct warming experiments that cover the range of expected temperatures.

In logistically challenging environments this is currently impossible using existing passive warming technologies (Marion et al., 1997). Here we have addressed this research need by designing and testing a novel Zero Power Warming (ZPW) chamber capable of unattended, power free, modulated temperature control of a vented enclosure



capable of elevating temperatures beyond those achievable with existing passive warming approaches. Our motivation for the development of this new warming method was improving understanding and model representation of photosynthesis and respiration in the Arctic tundra, but the approach described here could be applied to many different ecosystems.

## 2 Materials & Methods

The design goal for our ZPW chamber was to improve the range and regulation of the achievable temperature differential in a passively warmed chamber that can be operated unattended and without electric power. The novel aspect of this design is the use of vent actuation based on the temperature differentials of hydraulic reservoirs.

### 2.1 Vent actuation

Our aim was to design a chamber capable of maintaining a temperature differential between the inside and the outside of the chamber over a broad range of ambient temperatures. The basis for the ZPW vent control mechanism is the expansion of an incompressible fluid in response to increasing temperature. Since this is a linear response, we can design a system that provides a consistent volume difference at a given temperature differential. The absolute temperature range is limited by the physical properties of the fluid used. The fluid must remain a free flowing liquid throughout the expected temperature range, with a high coefficient of expansion and low vapour pressure. The system will not function correctly if the fluid solidifies or vapourises. For practical reasons, the fluid should be non-corrosive, readily available, low cost and non-toxic. For the prototype chamber we used a vegetable oil based hydraulic fluid (Hydro Safe® 130 VG 68, Hydro Safe, Inc., http://www.hydrosafe.com/) which has a low vapour pressure and stays in liquid form in the –10 to +20° C temperature range expected during the thaw season (June – September) at Barrow, AK.

Figure 1 illustrates the ZPW vent control mechanism. Under conditions where the temperatures inside and outside of the chamber are identical, the vent is closed (Fig. 1a). When the temperature inside the chamber is higher than the temperature outside the chamber, the vent will open (Fig. 1b). The extent of opening is proportional to the temperature differential between inside and outside of the chamber. The pistons in the cylinders respond to expansion and contraction of the liquid volumes in the heat exchangers due to changing liquid temperature, which follow the air temperature surrounding the heat exchange manifolds. The liquid volumes and piston positions are initially adjusted so the vent is closed when the temperatures inside and outside the chamber are equal (Fig. 1a). When the chamber interior is warmer than the ambient environment outside the chamber (Fig. 1b), the liquid within the heat exchanger located inside the chamber expands more than the liquid within the heat exchanger located outside of the chamber, causing the piston in cylinder B to extend more than the piston in cylinder E. This differential in piston extensions causes the vent (A) to open.

The heat exchangers were constructed from stainless steel to provide rapid heat exchange with the surrounding air and a high albedo to reduce direct heating from incident solar radiation. During early prototyping we found that copper



heat exchangers tarnish with time, reducing albedo, with the result that direct solar warming of the fluid caused erroneous vent actuation. The piston diameters and stroke lengths were selected to provide sufficient piston movement to actuate the vent across a broad ambient temperature range. Since the liquid volumes and cylinder diameters were the same, the piston positions in both cylinders were displaced equal distances when the absolute temperatures inside

and outside the chamber changed by the same amount. The piston cylinders were connected together so they moved as a unit. When both piston rods moved the same distance, the vent position did not change. Therefore, the vent position would only be affected by temperature differentials between the inside and the outside of the chamber, regardless of the absolute temperatures.

Each heat exchange manifold was equipped with valves and drain plugs to allow easy filling, draining and purging of gas bubbles during the initial setup and when changing the liquid. The manifolds were also equipped with pressure relief valves (Model # 2305C-400, Kepner Products Company, http://www.kepner.com/) to protect the hydraulic system from over pressurization if the pistons reach the cylinder end stops or the vent sticks in place. While not used in this prototype, a sliding connection could be installed where cylinder B connects to vent A (Fig. 1) to accommodate

negative pressure events when the temperature inside the chamber is significantly lower than the outside temperature. Although this condition can occur, we have seen this only rarely in our studies. In these instances, the flexibility present in the mechanism accommodated the resultant forces. The size, and thus the achievable temperature elevation range, of the ZPW vent control mechanism can be adjusted to match the needs of a particular experimental design and location.

**2.2 Chamber construction**

Our prototype chamber (Fig. 2) was sized to allow the study of low stature Arctic vegetation and provide easy access to the plants and monitoring equipment without having to remove the chamber from the research plot. The prototype chamber measured 2.4 m x 2.4 m x 1.4 m (L x W x maximum H). To minimise the chamber artefact of shading the vegetation and to maximize sunlight transmission, we designed this chamber to minimise structural components that

would block sunlight reaching the chamber interior. The chamber walls and roof were covered with F-Clean® (AGC Chemicals, www.agcchem.com), a 100 µm thick ethylene tetrafluoroethylene (ETFE) architectural glazing film that exhibits high light transmission throughout the solar spectrum, high structural strength, puncture resistance and low stretch, with little degradation in its optical or structural properties over time (Barton et al., 2010). The chamber framing was constructed using T-6061 alloy aluminium angle and stainless steel fasteners for mechanical strength,

corrosion resistance and to minimise structural shading. The vent panels were glazed with dual wall rigid polycarbonate sheeting (Macrolux® 10 mm twin wall, CO-EX Corporation, www.co-excorp.com).

**2.3 Evaluation of the prototype**

**2.3.1 Environmental Monitoring**

We compared the performance of a ZPW chamber with modulated venting to a control chamber with fixed venting

and an adjacent fully instrumented ambient plot. The chambers were deployed on the coastal tundra at the Barrow




Environmental Observatory (BEO), near Barrow, AK (71.3°N, 156.5°W; note that on 1 December 2016 Barrow was officially renamed Utqiaġvik following the original Inupiat name). The area we used for evaluating our passive warming approach was characterised by small thaw ponds and low- and high-centred polygons with a low diversity of vascular plant species diversity that was dominated by *Carex aquatilis* Wahlenb. The period of evaluation ran from June 15th 2016 to September 7th 2016 which covered the thaw season (Brown et al., 1980). We monitored: air temperature and relative humidity just above canopy height using a temperature and relative humidity sensor (083E-L, Campbell Scientific, Logan, UT, USA), soil temperature at 5, 10 and 15 cm below the base of the moss layer using a stainless steel temperature probe (109SS-L, Campbell Scientific, Logan, UT, USA), volumetric soil water content below the base of the moss layer with a soil moisture sensor (GS3, Decagon Devices, Pullman, WA, USA), solar radiation using a pyranometer (LI-200R, LI-COR, Lincoln, NE) and movement of the vents in the chambers using a string potentiometer (Model SP2-25, Celesco Transducer Products, Inc. Chatsworth, CA). Air temperature was also recorded every 15 minutes using standalone temperature loggers (UA-001-64, Onset Computer Corporation, Bourne, MA) as a backup (data not shown). The data from these instruments was collected once a minute over the 85-day evaluation period and stored on data loggers (CR-1000, Campbell Scientific, Logan, Utah) located adjacent to each chamber and the ambient plot. Hourly instrument measurement digests were wirelessly collected (via the standard 802.11 WiFi protocol) by a control computer in a nearby instrument hut and from there, transmitted back to Brookhaven National Laboratory via FTP where they were backed up on a server. These hourly instrument digests were parsed into summary figures and performance diagnostics using a custom script within the R environment (R Core Team, 2017) and served up via an external web site to provide near-real-time updates of experimental conditions and performance characteristics during the deployment. An Internet enabled camera (StarDot NetCam SC; StarDot Technologies, Buena Park, CA) was also used to monitor the experiment. All the instrument data are publically available (Lewin et al., 2016; Serbin et al., 2016).

### 2.3.2 Short term monitoring of carbon dioxide concentration

To evaluate the potential for draw down and accumulation of carbon dioxide in the chambers we used infrared gas analysers (LI-6400XT, LI-COR, Lincoln, NE, USA) to monitor carbon dioxide concentration ($[CO_2]$) in the air inside the ZPW chamber and in the ambient plot. The LI-6400s were zeroed at the field site with a common nitrogen standard (99.9998% nitrogen, $CO_2$ <0.5ppm, $H2O$ <0.5ppm; ALPHAGAZ 2, Air Liquide American Specialty Gases LLC) and measurements taken using an open leaf chamber with all environment controls turned off. The $[CO_2]$ was logged every 60 seconds for two days.

### 2.3.2 Performance of the chamber skin

To quantify the performance of the F-Clean® film, we measured the transmittance of brand-new film and film that had been deployed on an earlier prototype at Brookhaven National Laboratory for approximately one year. The transmissivity was quantified using a full-range (i.e. 350 to 2500 nm) spectroradiometer (HR-1024i, Spectra Vista Corporation, Poughkeepsie, NY) together with a fiber optic light guide attached to a measurement probe with an internal, full-spectrum calibrated light source. We measured the transmission by placing either a section of the new or





deployed film between the lens of the measurement probe and a 99.99% reflective Spectralon® standard (Labsphere, North Sutton, NH) to capture the energy transmitted through the film across each of the 1024 wavelengths measured by the instrument representing the visible (VIS), near-infrared (NIR), and shortwave infrared (SWIR) spectral regions. This allowed us to examine the possible reduction of solar energy into the chamber by the F-Clean® film across the

full shortwave spectral region, and the extent to which this was impacted by an extended field deployment. We also measured the first-surface reflectivity of the film material by placing a black absorbing material behind the film within the measurement probe.

### 3 Results & Discussion

#### 3.1 Chamber operational overview

Figure 3 shows a snapshot of data that illustrates how the ZPW functions. As solar radiation warms the air inside the ZPW chamber the piston attached to the internal heat exchanger begins to open the vents and prevents overheating, when solar radiation is lower or intermittent the vents remain closed or open only slightly. During this week all days had intermittent cloud cover except Day of Year (DOY) 203 (Fig. 3a). The resulting influence of solar radiation on ambient air temperature and the air temperature inside the ZPW can be seen in Fig. 3b. On these five days the air

temperature in the ZPW was clearly higher than the ambient air temperature, especially during the periods of high irradiance. On DOY 201 and 205 the incoming solar radiation was not as high as on DOY 202, 203 and 204 or highly variable due to intermittent cloud cover. Figure 3c shows how much the vents in the ZPW chamber opened during these days. On DOY 201 and 205 there was very little venting, whereas on days with higher irradiance and less cloud cover there was near continual active venting that peaked at solar noon. This differential venting enabled the ZPW

chamber to maintain a similar warming profile with respect to the ambient plot on days with varying solar radiation (Fig. 3d). Despite having very similar solar radiation profiles there were different venting profiles on DOY 202, 203 and 204 which likely reflects variable wind conditions (not measured) which would affect the exchange of warm chamber air with ambient air and hence the air temperature inside the ZPW chamber.

Figure 4 shows an additional four-day snapshot of data that included manipulation of the control chamber vent. Half way through the morning of DOY 190 we closed the vents on the control chamber (Fig. 4b) to better understand the impact of our modulated venting technology on the air temperature inside a chamber. On DOY 188 and 189 we see typical ZPW warming (Fig. 4c). On these days the ZPW chamber is warmer than the control chamber and solar radiation is sufficient to induce active venting (Fig. 4a & b). Following closure of the control chamber vent we see an

immediate spike in the control chamber air temperature, which rises to a peak of 13 °C above ambient. During the same day the ZPW chamber was venting and able to purge excess heat but still maintain a 4–5 °C temperature elevation above the ambient plot, thus enabling a modulated warming of air temperature. These data clearly demonstrate that ZPW chambers are capable of modulating chamber temperature passively (without the need for electric power) and can avoid overheating problems that can occur with closed or minimally vented chambers. Avoiding sustained high

temperature excursions on days associated with warmer temperatures and higher irradiance is essential for



experimental manipulations where such temperature excursions have the potential to terminate the experiment by damaging the vegetation inside the chamber or affect the experiment in other ways that could negatively affect the applicability of the results to real-world conditions.

## 3.2 Chamber warming

### 3.2.1 Air temperature

We collected data on the performance of the ZPW for most of the thaw season, which is the period when we would anticipate deploying passive warming in the tundra. Figure 5 shows the daily mean air temperature in the ambient plot, control chamber and ZPW chamber. Air temperature fluctuated markedly over the thaw season with temperatures

ranging from 0–15 °C. However, the air temperature inside both the control and ZPW chambers was consistently and significantly higher than the ambient air temperature ($t_{2,78}$, $P<0.001$, Fig. 5) and the daily mean temperature differential (2.6 °C) between the ZPW and the ambient plot was double the differential between the control chamber and the ambient plot (1.3 °C, $t_{2,78}$, $P<0.001$, Fig. 6).

It is not possible to conduct a meaningful direct comparison of the ZPW chamber performance with other Arctic passive warming approaches due to differences in location, size, seasonal variation in weather, the measurement location, use of shielded and unshielded thermocouples and the length of the trial period. However, the ZPW chamber had a warming effect that was double the adjacent partially enclosed, passively warmed control chamber with fixed vents and double the mean daily air temperature warming reported in other passive warming studies (Marion et al.,

1997; Welker et al., 2004; Jonsdottir et al., 2005; Wahren et al., 2005; Bokhorst et al., 2013). A recent global assessment of these passive warming approaches and Arctic greenhouse and infra heating experiments reported that in these experiments the mean daily summer air temperature was elevated by 1.5 °C (Elmendorf et al., 2012). Given that estimated increases in mean annual surface air temperature in the Arctic has been projected to be 2.5 °C by 2060 (ACIA, 2005), and for Alaska 1.9–3.1 °C by 2050 (Melillo et al., 2014), it is imperative that warming approaches,

such as the one presented here, are developed that are capable of greater temperature elevation.

### 3.2.2 Diurnal temperature range

The ZPW chamber, like any enclosure that relies on solar radiation for heating, has the potential to affect the diurnal temperature range. Both the control and the ZPW chambers had a significantly higher diurnal temperature range than the ambient plot as well as significantly higher daily temperature minima and maxima ($t_{(2),78}$, $P<0.001$, Fig. 7). On

average the daily minimum temperatures were 0.58 °C (control chamber) and 1.32 °C (ZPW chamber) greater than the average daily minimum temperature in the ambient plot. The temperature maxima were 2.50 °C (control chamber) and 4.43°C (ZPW chamber) greater than the average daily maximum temperature in the ambient plot. As expected, the increase in the diurnal temperature range was driven mostly by increases in the maximum temperature which were associated with days with high solar radiation.





### 3.2.2 Soil warming

Warming the air also warmed the soil. Figure 8 shows the impact of ambient solar radiation (Fig. 8a) and ambient air temperature (Fig. 8b) on the soil temperature differential at 5, 10 and 15 cm below the moss layer (Fig. 8c). The soil temperature at all three depths shows the same dynamic response that largely mirrors the pattern in air temperature and solar radiation. The difference in the soil temperature differential (with the ambient plot) between the ZPW chamber and the control chamber decreases with soil depth. Warming of the soil was greater than warming of air (Fig. 5, 6 & 8c). This likely reflects the fact that the soil acts as a heat sink for the elevated air temperature and is less influenced by air exchanges that will rapidly decrease the air temperature. This lag would expected as the soil takes longer to warm and is slower to cool than the air. However, we sited this prototype on the same footprint as a 2015 prototype (no data) and thaw and degradation of the permafrost that occurred in 2015 thickened the active layer which may have influenced soil temperature profiles in 2016.

### 3.3 Consideration of potential artefacts

All passive warming approaches have a chamber effect and the impact of chambers on plant growth has been considered in depth previously (Long et al., 2004; Marion et al., 1997). Whilst it is possible to effectively warm plants and ecosystems without the use of enclosures, alternative approaches such as infrared heating are often not practical in remote, logistically challenging locations. Therefore, if we want to understand how plants and ecosystems will respond to rising temperature in such locations we need to use passive warming approaches, but do so with a full understanding of their limitations. In short, selection of a passive warming approach requires a balance between the degree of warming and the potential artefacts, the greater the desired warming, the larger the potential for unwanted chamber effects. We have considered and quantified some of the artefacts associated the ZPW design. We have not considered all the issues that may be of importance to the broader community e.g. restricted access for pollinators and herbivores, but focused our attention on key variables that impact plant physiology i.e. light, vapour pressure deficit, $[CO_2]$ and soil moisture content.

### 3.3.1 Attenuation of solar radiation

The transmittance of the new and deployed F-Clean® film showed relatively stable values in the visible and near-infrared wavelengths, with minor reductions between 1630 and 1750 nm followed by marked reductions in the far SWIR wavelength region (Fig. 9). The average transmittance values of new film was 94% (±0.3%) and 95% (±0.5%) for the visible and NIR regions, respectively, while the transmittance values for film deployed for one year averaged 90% (±0.8%) and 92% (±0.5%). Wavelengths >2200 nm displayed a significant drop in transmittance (an increase in absorption by the film), with an average transmission of 74% (±8.5%) for the new and 71% (±8.2%) for the deployed film. Overall, the film exposed to the elements for one year displayed an average of a 4% (±0.8%) reduction in transmittance of shortwave radiation (i.e. 350 to 2500 nm) compared to unused film. We also found that the reflectivity of the film material was minimal, averaging 3% for both the new and deployed F-Clean® film. Fig. 9 also shows the solar spectral irradiance, comparing this spectrum with the transmission of the film clearly shows that the regions of maximum solar output correspond to the regions of highest transmission, including the important 400 to 700 nm PAR





region utilised in plant photosynthesis. The F-Clean® film performed considerably better than fiberglass and Lexan which have been typically used for passive warming chamber construction. Those materials have transmittance values of 86% (fiberglass) and 90% (Lexan) when new. To our knowledge, no data for the transmittance of these materials in deployed chambers has been reported (Molau & Mølgaard, 1996).

Measuring transmittance through the chamber walls, particularly of brand new materials, does not account for other sources of attenuation such as the chamber frames. Therefore, we measured solar radiation at canopy height inside the control and ZPW chambers and compared 1-minute readings to the solar radiation measured in the ambient plot. During the daytime (solar radiation > 5 W m$^{-2}$) the transmission of solar radiation inside the control and ZPW chambers

relative to the solar radiation measured in the ambient plot was 78% ± 15% SD and 76% ± 15% SD respectively. This ratio accounts for loss of light transmission associated with the chamber frames, the ZPW apparatus, aging of the film, variation in solar angle and potential dirt, condensation and raindrops. These transmission values are the true, and rarely reported, values for attenuation of solar radiation in our field enclosures.

### 3.3.2 Potential for reducing vapour pressure deficit

Any warming experiment will raise the vapour pressure deficit (VPD) of the air unless water vapour is added to the warmed air. However, the volume of potable water required for such an endeavour is enormous and the artefacts associated with maintaining VPD can be considerable (Hanson et al., 2017). As a result warming experiments rarely attempt to control VPD (Bronson et al., 2009). Elevation of VPD is a concern for plant physiology experiments if the warming treatment moves the VPD above 1.5 kPa as limitations on stomatal conductance and photosynthesis are

possible. In our prototype the mean daily relative humidity (RH) was lower in both the control chamber (88% RH) and the ZPW chamber (86% RH) compared to the ambient plot (96% RH). However, the average 9% reduction in RH in the ZPW chamber was physiologically insignificant in terms of VPD. Fig. 10 Shows the 1-minute data from the ambient plot, control chamber and ZPW chamber. During the period of study the VPD was only above 1.5 kPa on a couple occasions and the majority of the time the VPD was <0.5 kPa (Fig. 10). Therefore, the potential for negative

impacts of an elevated VPD on stomatal conductance and photosynthesis is minimal.

### 3.3.3 Potential carbon dioxide draw down and build up

When a plot is enclosed by a chamber there is potential for the vegetation and soil to influence the [$CO_2$] inside the enclosure, either through draw down of the [$CO_2$] through photosynthesis or elevation of the [$CO_2$] through respiration. Since the ZPW chambers are partially enclosed, we investigated the potential for a chamber effect on [$CO_2$]. We

monitored the ambient [$CO_2$] and the [$CO_2$] inside the ZPW over two days (Fig. 11). We found that during periods of high solar radiation when the ZPW chamber was actively venting the [$CO_2$] inside the chamber was ~4 µmol mol$^{-1}$ below ambient [$CO_2$]. When solar radiation dropped, and vents were typically closed, the [$CO_2$] inside the ZPW chamber rose by ~8 µmol mol$^{-1}$. At the start of the day on DOY 190 the vents were closed and as solar radiation increased the vegetation was able to draw down the [$CO_2$] inside the chamber by ~20 µmol mol$^{-1}$ before warming led

to venting and this larger [$CO_2$] differential vanished (Fig. 11). The effect of the enclosure on the internal [$CO_2$] at





this location was minimal and does not present a serious concern and is comparable to [$CO_2$] changes seen during still air conditions in other systems. However, the potential for altering the [$CO_2$] should be evaluated in other locations prior to initiating a full experiment. If changes in [$CO_2$] did present a problem they could be mitigated by tuning the system to maintain a permanent minimum venting, or through the addition of a small solar powered fan which could

be actuated by a [$CO_2$] sensor.

### 3.3.4 Soil water content

A major concern with any chamber system is exclusion of precipitation and dewfall. Even open topped chambers without frustums can exclude a significant amount of windblown precipitation. When coupled with a high rate of evapotranspiration there is significant potential to dry out the soil inside the enclosures. Our Arctic location presents

a best-case scenario for the use of partially closed field enclosures. The presence of permafrost and an active layer thickness of 20–70 cm results in a landscape that is poorly drained. Mean annual precipitation is only 114 mm but poor drainage coupled with low temperatures and high humidity means that evapotranspiration is very low and much of the landscape is covered by standing water (Brown et al., 1980;Shiklomanov et al., 2010).

Our measurements of volumetric soil water content measured in the centre of the ambient plot and the two chambers showed marked plot-to-plot variation that was not consistent with the warming treatment. This could reflect plot-to-plot variation in topography, but also the evolution of local drainage patterns as the thaw season progressed. A heavy rainstorm on DOY 191 demonstrated how quickly lateral flow can recharge the soil water content inside the chambers (Fig. 12). However, note that these data are confounded by the use of the same footprint in 2015 and 2016 as permafrost

thaw and degradation resulting from our treatment in 2015 may have altered local drainage patterns.

When considering the use of ZPW chambers in other systems exclusion of precipitation should be carefully considered. In wetlands or other locations with saturated soils, exclusion of precipitation may not be a major concern, but elsewhere external collection and immediate reapplication of precipitation into the enclosures would be an option

to mitigate this artefact (Nippert et al., 2009). Another option would be solar charged, battery-operated motors that could be used to open the vents in the event of a rainstorm, the opposite of approaches used to deploy rainfall exclusion treatments (Gray et al., 2016). For short-term deployments, for example to investigate enhanced heat wave effects, the exclusion of precipitation might not be an issue. However, we strongly recommend monitoring soil water content and providing daily reports that can be used to inform potential manual watering efforts.

### 4 Conclusion

Here we demonstrate the successful design, construction and testing of a novel hydraulic system capable of elevating and modulating the temperature inside a passively warmed field enclosure. Our design was able to raise the mean daily air temperature to 2.6 °C above an adjacent ambient plot, twice the temperature elevation attained in a control chamber that lacked our hydraulic system. Our fully autonomous chambers required no power and were able to avoid

high temperature excursions that occur in fully, or near fully, enclosed chambers. The design of the ZPW chambers is



highly flexible and can be adjusted to meet research needs in a wide range of logistically challenging environments, including tuning of the venting system to attain higher temperature differentials. This advance opens up the possibility for markedly higher warming in remote and logistically challenging environments and also the ability to conduct short-term heat stress experiments.

As discussed above there are many different approaches that can be used to understand the response and acclimation of plants and ecosystems to the warming. All of these approaches have advantages and drawbacks and all approaches have artefacts. The influence of potential artefacts on the ability to address a specific hypothesis need to be carefully considered, and should be used to determine the manipulative approach most suitable for a given scientific question.

**5 Acknowledgements**

This work was supported in part by The Next Generation Ecosystem Experiments Arctic project that is supported by the Office of Biological and Environmental Research in the Department of Energy, Office of Science, and through the United States Department of Energy contract number DE-SC0012704 to Brookhaven National Laboratory.

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





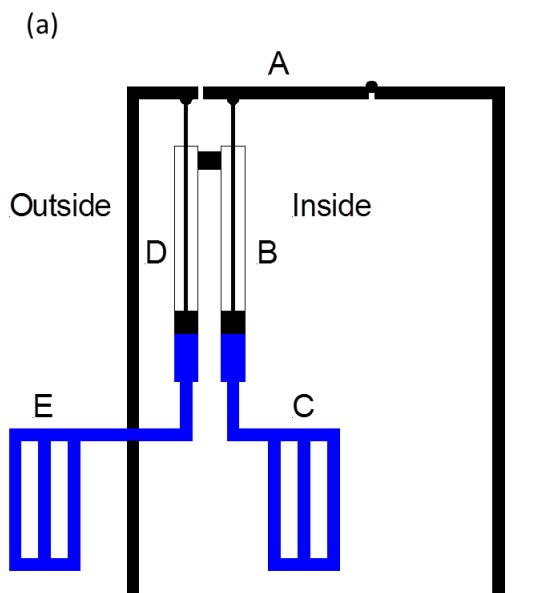
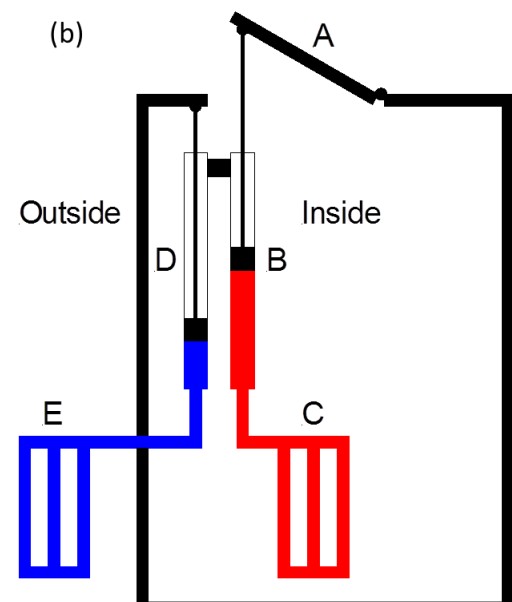

**Figure 1** Schematic diagrams illustrating how the vent control system responds to air temperature differentials between the inside and outside of the chamber. Movable vent (A) is connected to a hydraulic cylinder (B) which is connected to a liquid-filled heat exchanger (C) located inside the chamber. Cylinder B is connected to another hydraulic cylinder (D). A pivoting bar (not shown) extends from the connecting link to the chamber frame to stabilise the pistons. A counter weight connected to the connecting link applies upward force on both cylinders to counteract their weight and maintain positive internal pressures on both cylinders. The piston rod on cylinder D is connected to a fixed point on the chamber and hydraulically connected to another liquid-filled heat exchanger located outside the chamber (E). The pistons in the cylinders respond to expansion and contraction of the liquid volumes in the heat exchangers due to temperature. The liquid volumes and piston positions are initially adjusted so the vent is closed when the temperatures inside and outside of the chamber are equal (Panel a). When the chamber interior is warmer than the ambient environment outside the chamber (Panel b), the liquid in the heat exchanger inside the chamber (C) will expand more than the fluid in the heat exchanger outside the chamber (E), causing the piston in cylinder B to move more than the piston in cylinder D. This differential in piston positions causes the vent (A) to open.



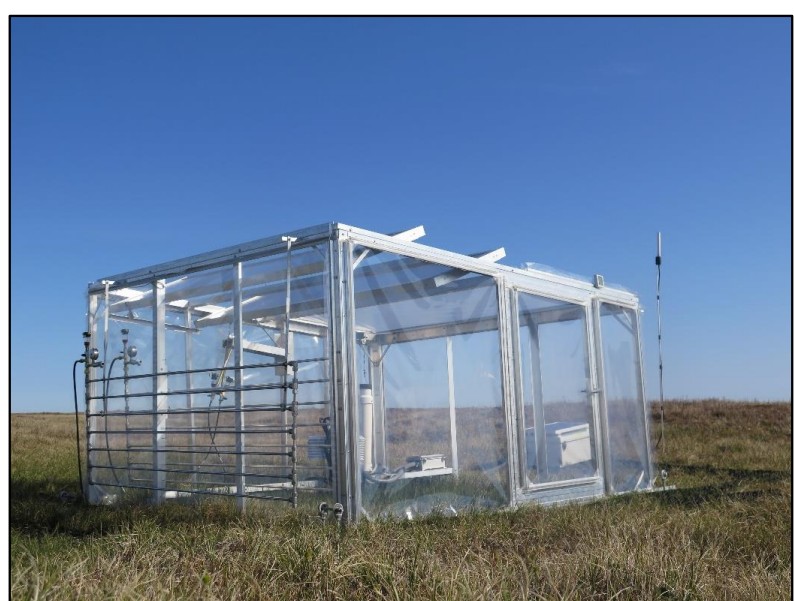

**Figure 2** The Zero Power Warming (ZPW) chamber pictured on coastal tundra near Barrow, AK. The chamber measures 2.4 m x 2.4 m x 1.4 m (L x W x maximum H). In this picture, the two vents are partially open due to the temperature differential between the internal and external heat exchangers that are visible on the near wall of the chamber. The brass pistons and counter weight (section of white PVC pipe) can be seen opposite the door. The near corner obscures the view of most of the monitoring instrumentation but the junction box can be seen in the center of the far wall. Outside and behind the chamber you can see a box with a solar panel that contains a data logger and battery to power the monitoring instrumentation, and an antenna for data communication.





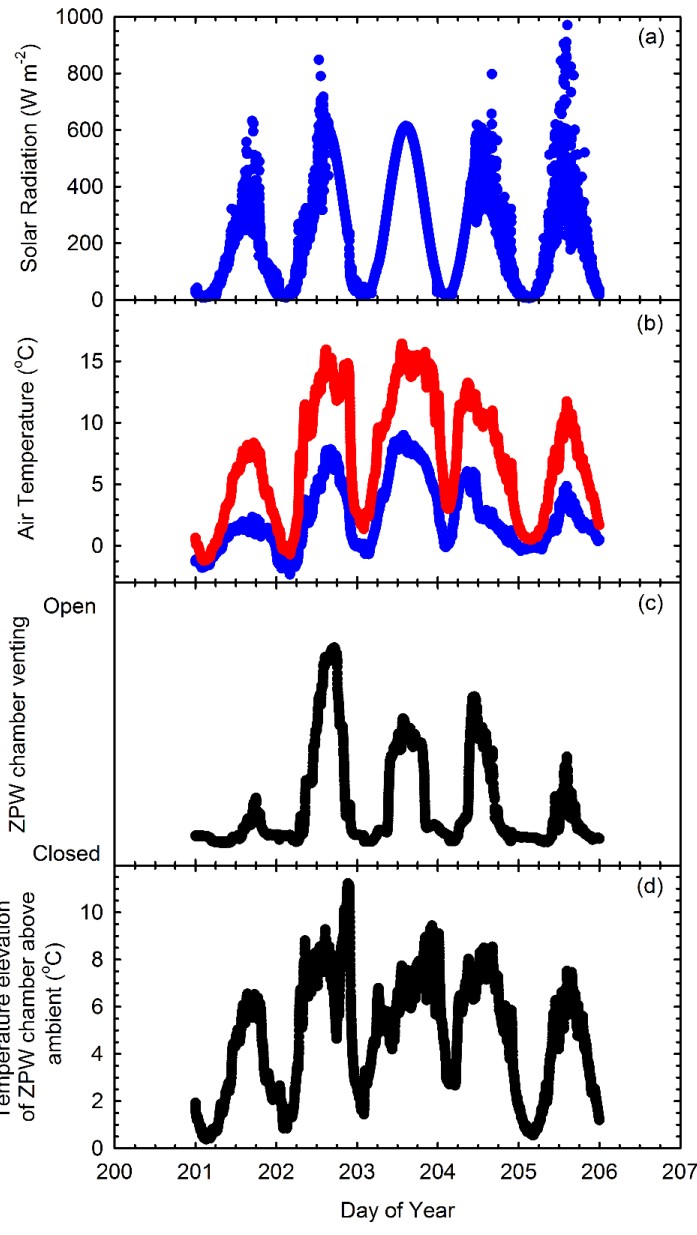

**Figure 3** A snapshot of ZPW performance. Energy for warming comes from ambient solar radiation shown in panel a. Panel b shows the air temperature in the ambient plot (blue) and inside ZPW chamber (red). Panel c shows the degree to which the vents opened due to the temperature differential between the internal and external heat exchangers as measured by a string potentiometer connected to the edge of the vent. Panel d shows the air temperature differential between the ZPW and the ambient plot. Plots show 1-minute data.





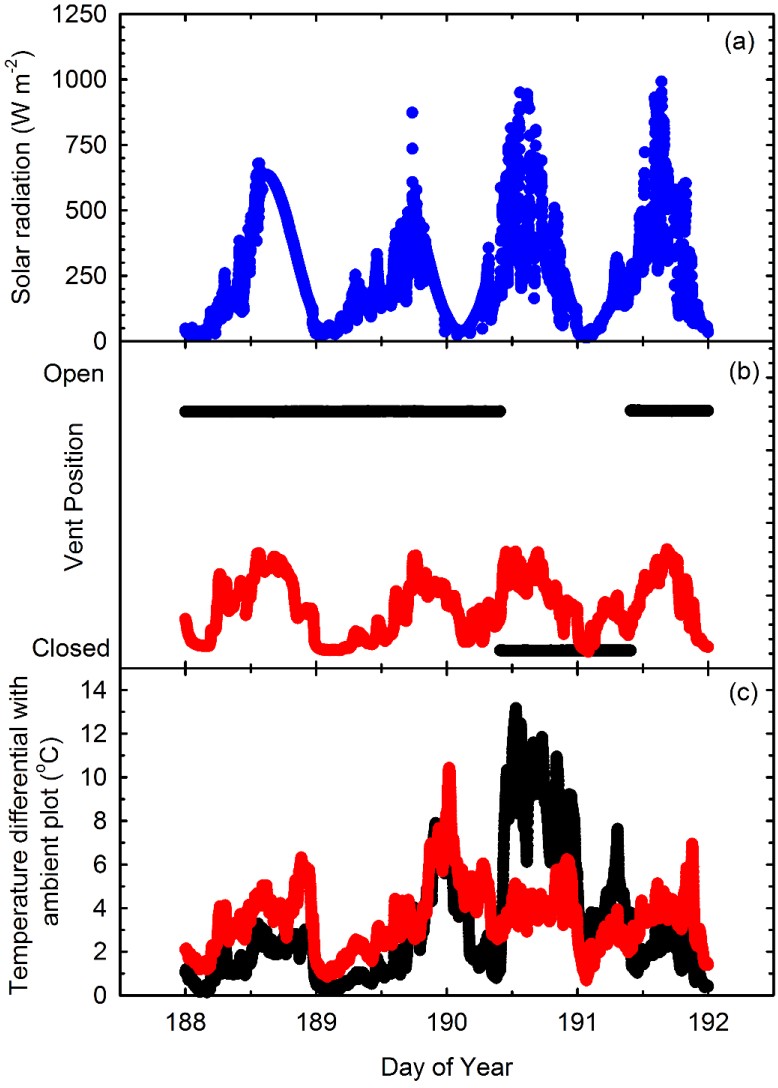

**Figure 4** A four-day snapshot of data demonstrating how the ZPW chamber can modulate the air temperature inside a chamber. Panel a shows ambient solar radiation. Panel b shows the degree of active venting by the ZPW chamber (red symbols) and the position of the fixed vent on the control chamber (black). Panel c shows the air temperature differential between the ZPW chamber and the ambient plot (red) and the control chamber and the ambient plot (black).





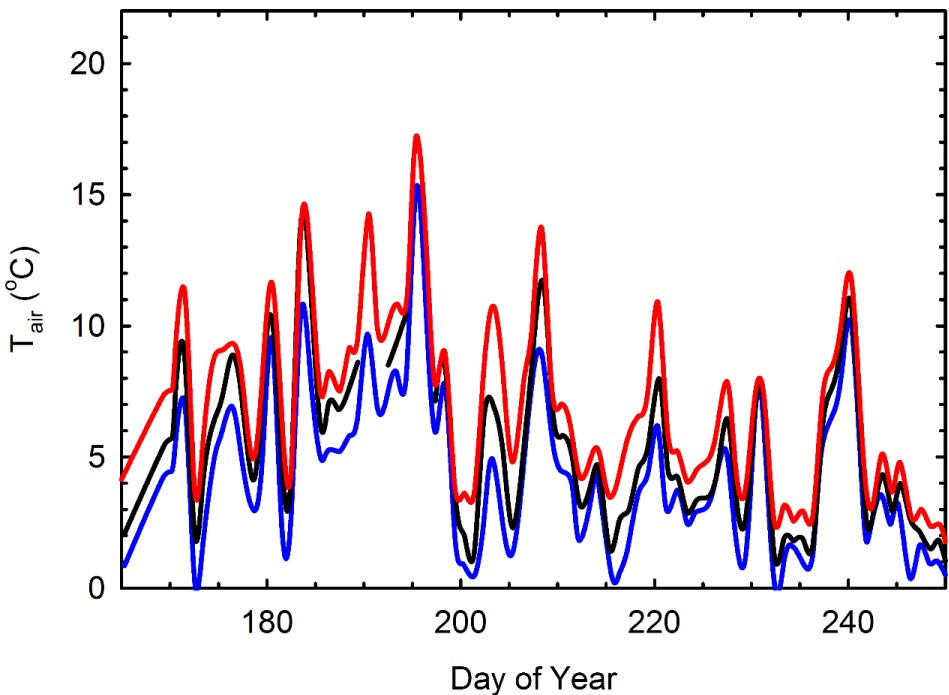

**Figure 5** Mean daily air temperature measured at canopy height in the ambient plot (blue line), the control chamber, with fixed venting (black line), and the ZPW chamber, with modulated venting (red line). In order to evaluate chamber effects the control chamber was closed on DOY 190, 191, 195 and 196; data for these days are not shown.





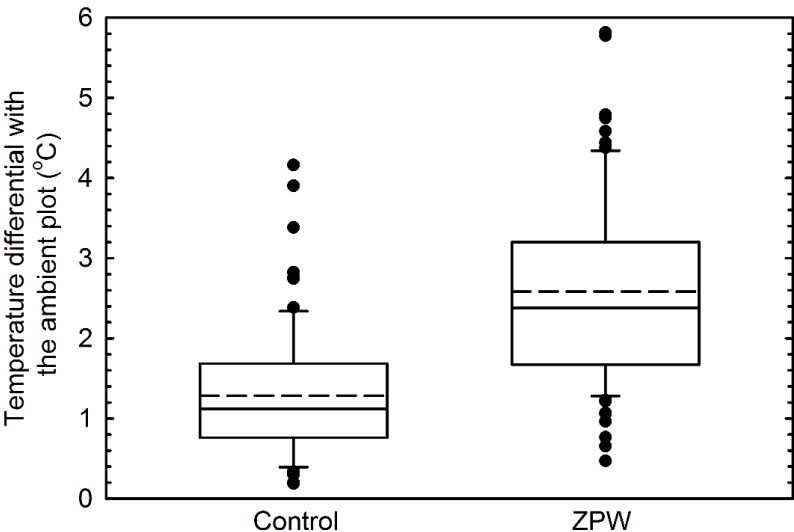

5 **Figure 6** Tukey box plots showing the temperature differential of the mean daily air temperature between the control chamber and the ambient plot and between the ZPW chamber and the ambient plot. Warming in the ZPW chamber was double the control chamber. Box plots show the interquartile range (box), median (solid line) and mean (broken line). The whiskers show lowest and highest datum still within 1.5 x inter quartile range of the lower and upper quartiles. Outliers are shown as black dots.





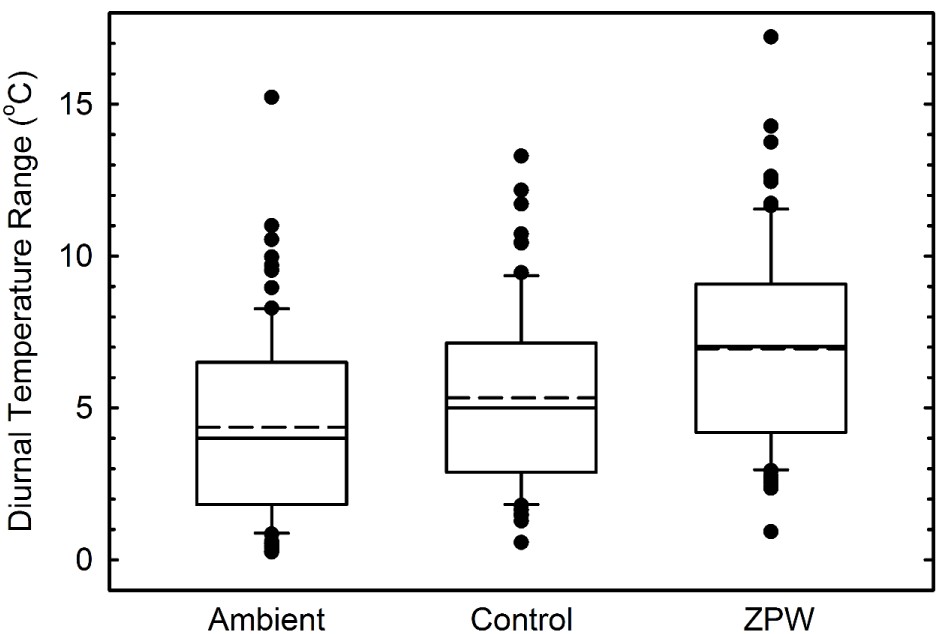

**Figure 7** Tukey box plots showing the mean daily diurnal temperature range in the ambient plot and the control and ZPW chambers.

5   The diurnal temperature range was calculated for each day by subtracting the minimum temperature from the maximum temperature recorded on a given day. Data from DOY 190, 191, 195 and 196 were omitted due to chamber manipulations. Box plots show the interquartile range (box), median (solid line) and mean (broken line). The whiskers show lowest and highest datum still within 1.5 x inter quartile range of the lower and upper quartiles. Outliers are shown as black dots.





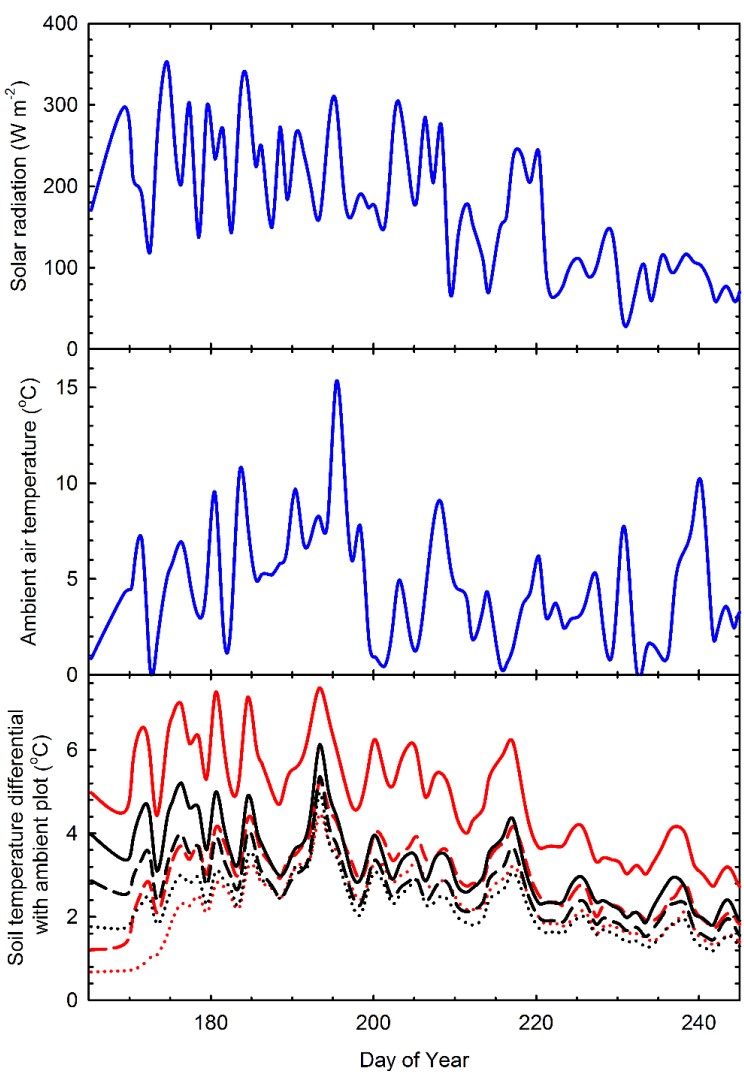

5  **Figure 8** The influence of ambient (blue) solar radiation (panel a) and ambient air temperature (panel b) on soil temperature in the control (black, panel c) and ZPW chambers (red, panel c). Panel c shows the temperature differential between the chambers and the ambient plot. Soil temperature was measured at 5 cm (solid line), 10 cm (broken line) and 15 cm (dotted line) below the base of the moss layer.





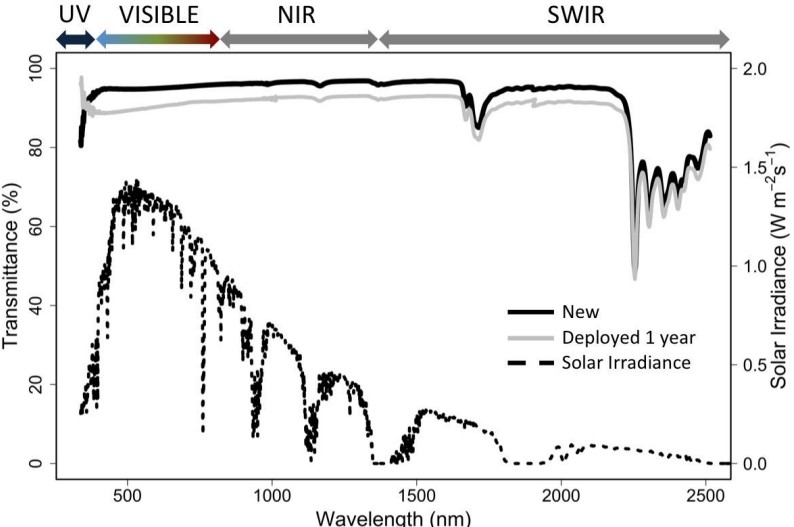

**Figure 9** The full short-wave (350 to 2500 nm) transmittance of the F-Clean® film material. The two solid lines depict the measured transmittance through brand-new film (black) and film deployed for one full year (grey). To better understand the impact of the film on transmission of solar energy into the ZPW chamber, we also provide the American Society for Testing and Materials (ASTM) Reference Solar Spectral Irradiance for an absolute air mass of 1.5 (black dashed line, http://rredc.nrel.gov/solar/spectra/am1.5/) for the same spectral region.





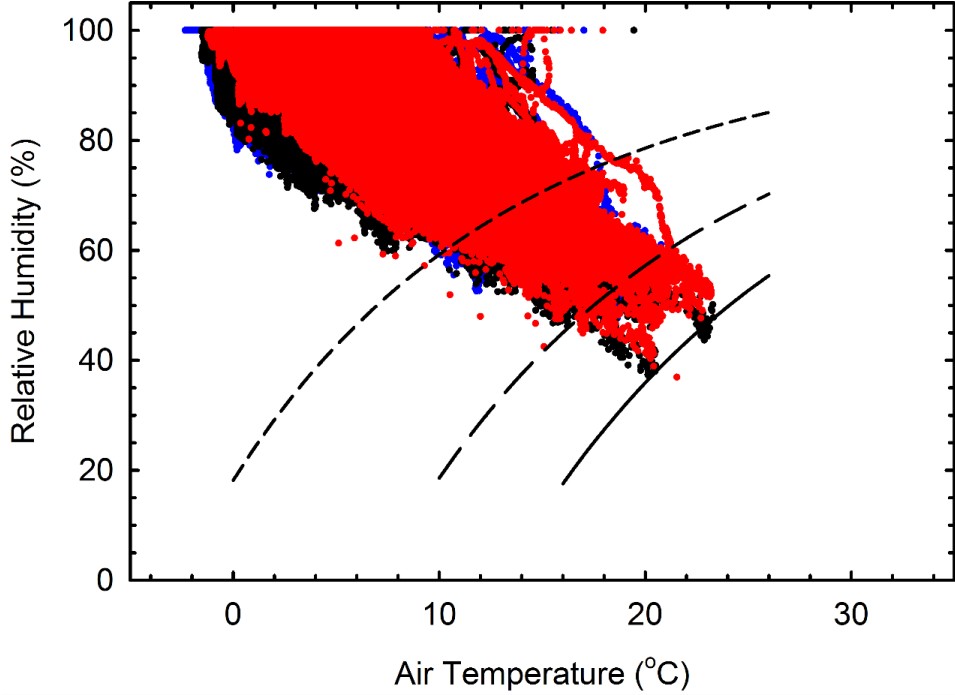

**Figure 10** One minute readings of relative humidity plotted against air temperature measured in the ambient plot (blue), the control chamber (black) and the ZPW chamber (red) over our trial period. The vapour pressure deficit is shown for 0.5 kPa (broken line – short dashes), 1.0 kPa (broken line – long dashes) and 1.5 kPa (solid line). Data from DOY 190, 191, 195 and 196 were excluded due to chamber manipulations.





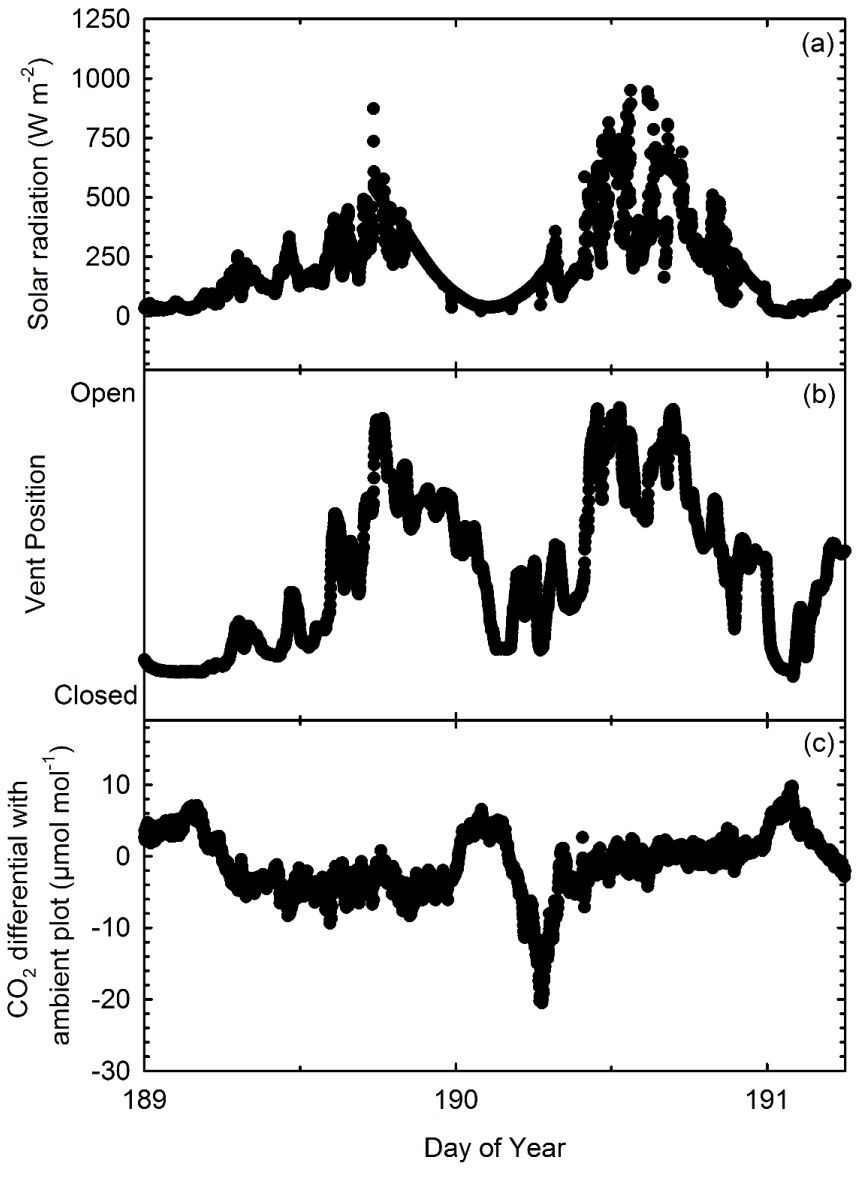

**Figure 11** The influence of solar radiation (panel a) and venting of the ZPW chamber (panel b) on the carbon dioxide concentration differential between the inside of the ZPW chamber and the ambient plot (panel c).





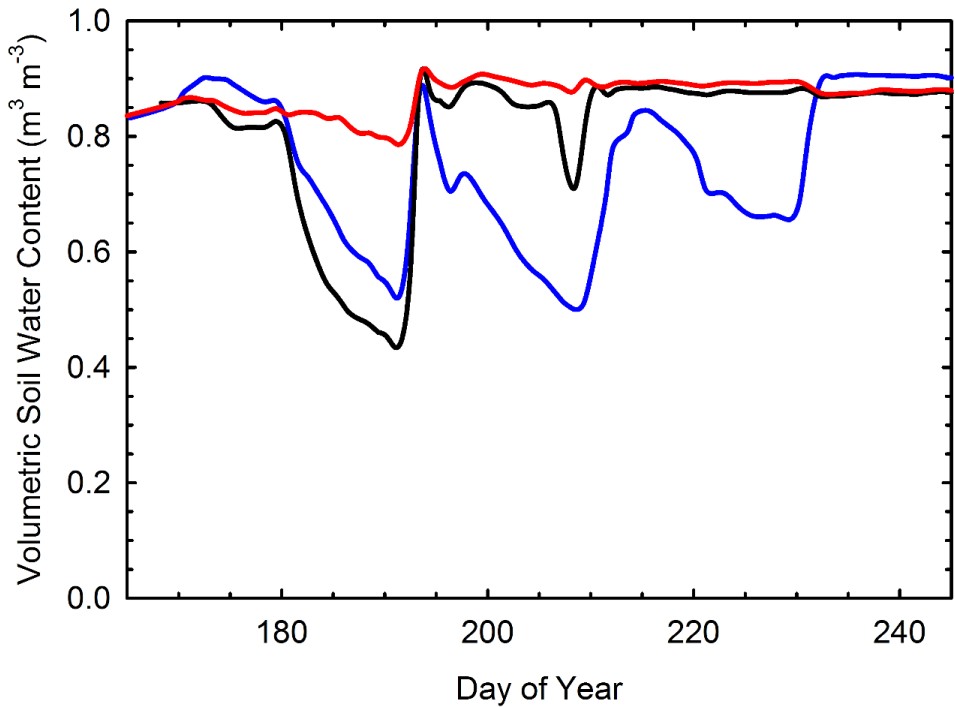

**Figure 12** Volumetric soil water content measured in the centre of the ambient plot (blue), the control chamber (black) and the ZPW chamber (red).

