# Peer review of "A zero power warming chamber for investigating plant responses to rising temperature"

_Biogeosciences, 2017_

## Referee Comment (RC1) · Anonymous Referee #1 · 7 Jul 2017

This is a well prepared and easy to read summary of a new method for partial control of a passive solar warming chamber. The ability to modulate (to a degree) the warming from solar-induced heating of the chambers provide some control over the nature of the warming treatments. Limited and quantified artifacts related to relative humidity, vapor pressure deficit and atmospheric $CO_2$ concentrations are openly discussed for the reader. I think the paper will be well received by the experimental warming community. This new approach will provide another tool for researchers to consider as they plan manipulative warming efforts in remote areas.

The following issues should be addressed by the authors prior to final publication.

Page 1 Line 20: Rather than quoting the daily mean temperature I recommend that the abstract quote both the daytime and nighttime mean temperature differentials. They

are not the same, and that reality should be stated up front. The discussion of diurnal temperature ranges on Page 7 provides the necessary text. It may also be useful to include a statement on the impact on soil warming in the abstract as well. Although it is stated in the text the abstract is not clear that these data are for the growing or thaw season. That may not be obvious to non-arctic readers of the paper.

Page 2 Line 37: Please consider removing the word "control" from the phrase modulated temperature control". Warming in the ZPW chambers is driven by solar inputs, and the venting process doesn't really control warming levels in the ZPW chambers. I like the authors' term modulation and how it is used throughout the manuscript. A reader who takes a quick look at the paper and its conclusions, should not think that the ZPW system specifically controls the internal temperature in the sense of a fully active warming system. The wording on page 6 lines 32 and 33 captures the ZPW process in acceptable terms.

Page 8 Line 1: I would use "influence" in place of impact.

Figures:

It would be helpful if the use of colors for various variables and chambers were the same in all plots. For example, red lines in Figures 3 and 4 represent different things.

Figure 8 lower box would benefit from a legend for the various lines. They are defined in the text, but a legend would be helpful.

Figure 10 doesn't work for me. I would replace it with a stacked set of frequency distributions of the temperatures for the ambient, control and ZPW chambers so that details of any skewness with respect to temperature can be directly observed without being obscured by the data printed on the top.

---

## Referee Comment (RC2) · Anonymous Referee #2 · 10 Jul 2017

In this manuscript [bg-2017-208] Lewin et al. describe a novel chamber design for terrestrial ecosystem warming experiments. The chamber utilizes heat exchangers and a hydraulic piston system to vent a greenhouse-like chamber. Since this system operates on relatively simple, mechanical principles and does not require electrical power, it offers a potentially useful tool for experiments in remote settings where power supply and maintenance are often major challenges. The authors demonstrate that the prototype chamber increased daytime air temperature and average of 2.6 C, compared to an ambient plot and could achieve higher temperatures than a reference chamber with a fixed level of venting while avoiding the extremely high temperatures of an unvented chamber. As the authors rightly point out, there is a need for expanded experimental studies on the effects of warming temperatures on plants and ecosystems. Further-

more, they note that all warming methodologies have potential shortcomings and that new warming techniques may be required for certain geographical, ecological (and cost) considerations. Overall, the authors present convincing evidence that this prototype chamber performs well under favorable, arctic growing season conditions. I list my questions and concerns below.

General Comments

I would also like to see more explanation of how the system is adjusted to control internal temperature (e.g. page 3, lines 27-32). It sounds like by adjusting the pistons, fluid, and vents, the chamber could be adjusted to maintain either a higher or lower amount of warming, specific to the local conditions. If so, this is a nice feature that could be highlighted more, but it also sounds like it might also require a lot of work during the installation phase.

Since the authors' intended setting for this equipment is in the arctic, it would be useful to know more about how the system might cope with more adverse weather conditions. Will the system be damaged if the hydraulic fluid freezes (below -10C) and if so, would the system need to be taken down before colder temperatures are expected? Additionally, how does the chamber perform under high winds or snow?

Overall, I found the design quite clever and am convinced that it performs better than a similar chamber with a constant amount of venting. However, I'm curious to know why they chose this approach, rather than a slightly more complex system, such as a computer-controlled venting system. Given that there is likely some power being used at one of these experimental sites (to operate meteorological dataloggers), and that solar and battery technology is increasingly efficient, adding a simple computer control system could be programmed to control temperature more precisely (possibly activating hydraulically-activated vents). The system relies on passive solar, so my guess is that solar panels might work. Cost and transport considerations could be the key factors, but I didn't see a clear rationale. Perhaps providing some information on shipping

weight and volume or cost comparisons could help to really show the advantage of this method. Otherwise, if it's simply a matter of reliability, that's potentially valid as well.

Specific Comments

From an organizational standpoint, I would consider the analysis of attenuation of solar radiation is a characteristic of the "chamber operational overview" and should appear in that section (3.1). Especially considering the prototype and reference chambers had similar materials.

The suggestion that the "potential for negative impacts of an elevated VPD on stomatal conductance and photosynthesis is minimal" (P9, L24-25) may be valid for this system, it may be worth noting that it may not be valid for all ecosystems.

Technical Corrections

The term "snapshot" (P6, L10 and elsewhere) seems a bit colloquial. Consider using a phrase like "sample time series" or "sample period of data".

---

## Author Comment (AC1) · 24 Jul 2017

Referee #1

Most of the data reported from passive warming experiments is daily mean temperature and therefore we provide this data to facilitate comparison with other approaches. The reviewer raises an important point about presenting daytime and nighttime mean temperature differentials. In many locations this would be a important result to report. However, in Barrow Alaska, there was no "night" during most of the time period we ran this experiment. We could have selected an arbitrary solar radiation threshold to define day and "night" but low levels of solar radiation were also observed under heavy cloud cover. Therefore, as noted by the referee, we framed our discussion around the diur-

nal temperature range. The referee also raises some good points about how readers would interpret our data given that it was collected in an Arctic system. We agree that it would help the reader if we emphasized some of the unique features of the Arctic e.g. 24h daylight, permafrost.

We agree with the suggested focus on modulation rather than control and would be happy to make that change in addition to other suggested edits.

Figures

We have attempted to use color consistently throughout the manuscript. In all cases we aimed to use red when discussing the ZPW chambers, black for the control chamber and blue for ambient. Where we discuss a differential between ZPW and ambient that differential is also shown in red and differentials between the control chamber and ambient are shown in black. We did spot a couple of discrepancies that could be changed for consistency throughout the manuscript; in figure 3 the plots in panels c and d could be in red and in figure 11 the plots in panel a could be blue and c and d could be red. The referee's specific point is possibly due to the use of red and black to distinguish differential temperatures in addition to absolute temperatures. We feel that introducing two additional colors would not enhance understanding.

We would be happy to amend the legend for Fig 8c.

When viewed independently from the text the goal of figure 10 is not clear and therefore we'd be happy to add some text to the legend to make our focus on VPD clear. The goal of figure 10 is to demonstrate that the VPD very rarely increased to the point where it might be expected to negatively impact stomatal conductance (>1.5 kPa). We feel our point is clearly demonstrated by Figure 10 but agree that three panels (one each for ambient, control and ZPW) may allow the reader to better assess the data. However, we disagree with the referee's suggested temperature frequency plots because a frequency distribution of temperatures would not communicate the same information and is a topic we covered earlier in the manuscript in figure 5 through 7.

Referee #2

General comments

The referee is correct that the design can be tuned to a specific environment and a desired amount of warming. While the paper does not describe this in detail, it is quite simple to tune the chambers to raise or lower the temperature differential and both procedures outlined below can be accomplished by one person within a few minutes. However, it is important to note that tuning requires access to data on the air temperature inside and outside the chamber and the result of any adjustment has to be monitored over time to determine the effect of the tuning on the magnitude and rate of warming.

(1) The first, and most straight forward method to change the magnitude of warming, i.e. the temperature differential between the inside and outside of the chamber, is to adjust the relative extensions of the pistons connected to the internal and external heat exchangers. This changes the "preload" on the vents, which affects how quickly the vents will begin to open as the temperature differential increases. If the preload is reduced, the vents will not immediately begin to open when the temperature differential becomes positive. If the preload is increased, the vents will begin moving as soon as the temperature differential becomes positive. This will increase the degree of vent opening for a given temperature differential, lowering the maximum differential. Adjusting the preload is easy to do and requires no tools. Simply open a valve to one of two oil reservoirs in the interior heat exchanger assembly (we have one on the piston and one on the heat exchanger), lift the vents open, which draws more oil into the cylinder as the piston rod is extended, and then close the valve to the reservoir. Similarly, oil can be pushed out of the system into a reservoir by opening the valve and lowering the vents, thereby reducing the preload and increasing the temperature differential required to initiate vent opening. (2) The sensitivity of the system to a temperature differential can also be adjusted by changing the attachment point where the piston rod is connected to the vent. We drilled a set of holes to allow us to easily adjust the distance

of the piston attachment point to the fulcrum of the vents. Moving the attachment point closer to the fulcrum results in greater vent opening and more rapid cooling for a given change in temperature differential and piston movement. Moving the attachment point further away from the fulcrum makes the vents less responsive.

Our initial prototype chamber was operated for three years at Brookhaven National Laboratory on Long Island, NY. It weathered tropical storms Irene and Sandy and withstood significant winter snow storms. Our Arctic prototype withstood deployment during the thaw season, including winds in excess of 20 m/s. In Barrow we anchored the chambers to the ground by sinking metal rods into holes drilled into the permafrost, allowed them to freeze in place and then attached the chamber to the anchors. We have no plans to leave the chambers out through the Arctic winter, but expect the structures could survive the low temperatures, high winds and snow loads. The heat exchanger settings and fluid would need to be changed to accommodate the lower temperatures.

There are many potential options for the fluid used in the heat exchangers. We decided to use hydraulic oil because it has a relatively large coefficient of heat expansion and low vapor pressure. We chose a vegetable based hydraulic fluid due to its low toxicity and biodegradability. The hydraulic fluid used in this study has a pour point below $-20°C$. There are similar hydraulic fluids available with pour points down to $-54°C$. Deployment at alternative locations may require the use of heat exchanger fluids with different properties.

We acknowledge the recent advances in solar and wind power options highlighted by the reviewer. Our group does in fact develop and operate equipment and facilities that use control systems powered by solar, wind and grid based electricity. These systems usually work well, but there can be issues with their long-term reliability, especially in remote locations and harsh environments. Our goal for this study was to design a system that did not require any electric power so that the chambers could be deployed with no supporting infrastructure and contain a minimal number of potential points of failure. While we used a relatively extensive and complex collection of sensors,

data loggers and communications equipment to document the performance of these chambers for this report, the bare minimum equipment required for an experiment is a chamber equipped with a battery powered temperature recorder. Of course, in a location with reliable electric power you could install a complex electronic control system which includes precipitation activated opening of the entire roof, infrared gas analyzers to modulate $CO_2$ concentration, supplemental heating for nighttime and other periods with low solar inputs and other desirable features. Such a system has recognized advantages, but each of these additional features brings with it additional complexity and additional points of failure. In harsh environments remote from technical support, there are advantages to reduced complexity.

Since the concept we describe here could be applied to a wide range of chamber dimensions and tailored to individual research needs, the shipping weight, size and costs are specific to the size of the (dismantled) chambers, the number of chambers, and the ship to and ship from addresses.

Specific comments & technical corrections

We agree with the referee's comments and would be happy to make these changes.

---

## Author Response (AR1)

**Changes to the manuscript and response to points raised during review**

The manuscript was improved following the suggested changes of referees 1 and 2 and our treatment of their comments closely follows our posted author response. These changes can be readily identified in the manuaction of the man

the marked up version of the manuscript and major changes are highlighted below

**Referee 1**

5

10

We added text to the abstract to emphasize that the data set was taken during the thaw season and added detail in section 2.3.1 to highlight some of the unique features of the Arctic.

We replaced the use of control with modulate when discussion temperature manipulation in the ZPW chamber

We updated our figure set to ensure consistent use of color through the manuscript, specifically figure 3 and Figure 11

After looking at figure 8c again we decided not to add a key to the panel. The detail is already contained in the figure legend and an in-panel-key would require us to show all six possibilities because of the

15 confusion associated with any color choice used for the key describing the solid, broken and dotted lines.

We broke figure 10 into three separate panels as described in our author response.

**Referee 2**

We have added a paragraph on tuning at the end of section 2.1

20 We added some text on the hydraulic fluid choice at the end of the first paragraph in section 2.1

We added some additional text to our conclusion to address the issue of powered venting approaches

On further review we decided that moving the section on attenuation of solar radiation was not a good idea and that it works best in its current position. This section is not just a description of the film but also details total solar radiation and compares transmission with other materials commonly used for

25 passive warming. Therefore it is more suited to the results and discussion section rather than the suggested methods section.

We added some text to the conclusion to address the referee's comment about deployment in other environments and also a few words at the end of section 3.2.2

1

The term snapshot was replaced throughout the manuscript

[revised manuscript text omitted]